# ALR$^2$: A Retrieve-then-Reason Framework for Long-context Question Answering

## Abstract

The context window of large language models (LLMs) has been extended significantly in recent years. However, while the context length that the LLM can process has grown, the capability of the model to accurately reason over that context degrades noticeably. This occurs because modern LLMs often become overwhelmed by the vast amount of information in the context; when answering questions, the model must identify and reason over relevant evidence sparsely distributed throughout the text. To alleviate the challenge of long-context reasoning, we develop a retrieve-then-reason framework, enabling LLMs to reason over relevant evidence collected during an intermediate retrieval step. We find that modern LLMs struggle to accurately retrieve relevant facts and instead, often hallucinate *"retrieved facts"*, resulting in flawed reasoning and the production of incorrect answers. To address these issues, we introduce ALR$^2$, a method that augments the long-context reasoning capability of LLMs via an explicit two-stage procedure, i.e., aligning LLMs with the objectives of both retrieval and reasoning. We demonstrate the efficacy of ALR$^2$ for mitigating performance degradation in long-context reasoning tasks. Through extensive experiments on long-context QA benchmarks, we find our method to outperform competitive baselines by large margins, achieving at least 8.4 and 7.9 EM gains on the long-context versions of HotpotQA and SQuAD datasets, respectively.

## 1 Introduction

The ability for large language models (LLMs) to reason over long contexts is critical in many downstream tasks, e.g., document analysis (Wang et al., 2024a), multi-hop tool use (Yao et al., 2023), agents with long history (Park et al., 2023), etc. Significant efforts have been made to extend the effective context length of LLMs, ranging from investigating how different modules in Transformer (Vaswani, 2017) architecture impact long-context performance (Su et al., 2024a; Chiang & Cholak, 2022; Xiao et al., 2024) to proposing new architectures with improved efficiency (Dai et al., 2019; Beltagy et al., 2020; Chevalier et al., 2023).

While these developments are promising, in our preliminary study, we show that the long-context performance of LLMs varied significantly across different tasks. In particular, we design a series of experiments to evaluate the performance of LLMs on long-context retrieval[1] and reasoning (Figure 1). We observe that, when tasked to generate answers by *directly* reasoning over the full context, performance degrades as the input context grows. In contrast, when tasked with retrieving the set of evidence relevant to the question, the performance of LLMs is only mildly affected by the growth of the input context.

Those findings motivate us to explicitly leverage an intermediate retrieval step to address the challenge of LLMs in long-context reasoning. In our proposed retrieve-then-reason framework, an LLM functions as both a retriever and a reasoner. When tasked to answer a question, it first collects the relevant evidence sparsely distributed across the context, and then derives the final answer by reasoning over that collected evidence. We also find that modern LLMs struggle to accurately retrieve "relevant facts", i.e., simply prompting them to retrieve supporting evidence often results in severe

---

[1]Following the definition in prior work (Chen et al., 2023; Hsieh et al., 2024), i.e., retrieval is performed by generating the relevant information from a given input text.

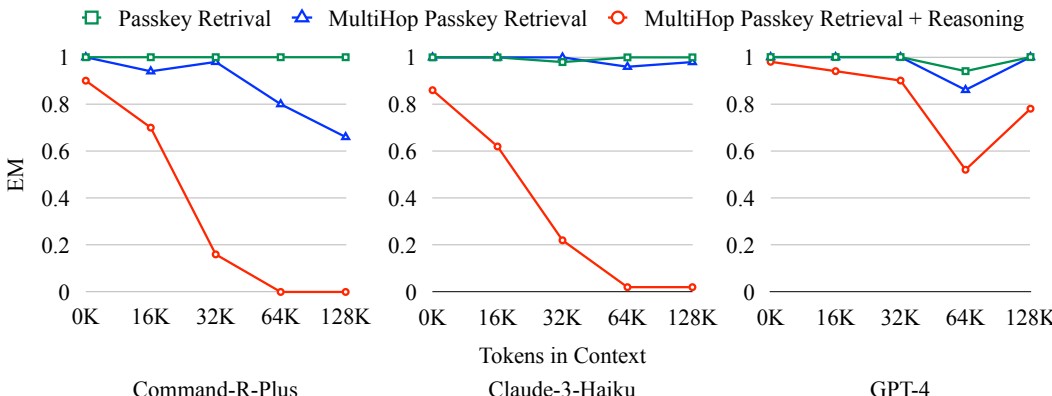

Figure 1: Performance of LLMs on three increasingly challenging long-context tasks. The x-axis represents the number of tokens in the long context, while the y-axis indicates the exact match score.

hallucinations, which leads to flawed reasoning by the LLMs caused by the reliance on inaccurate information. To this end, we introduce the $ALR^2$ approach, which augments the long-context reasoning capability of LLMs via an explicit two-stage procedure, i.e., ALigning LLMs with the objectives of both Retrieval and Reasoning.

Extensive experiments on long-context single-hop and multi-hop question answering tasks demonstrate the effectiveness of our $ALR^2$ approach. We compare $ALR^2$ against various advanced prompting techniques, including direct-answering and other popular prompting methods for long-context reasoning (Weston & Sukhbaatar, 2023; Lee et al., 2024). Results show that $ALR^2$ outperforms these approaches by at least 23.4 and 12.7 EM scores on the long-context versions of HotpotQA (Yang et al., 2018) and SQuAD (Rajpurkar et al., 2016) benchmarks. In addition, we also compare $ALR^2$ with LLMs fine-tuned with direct-answering prompting and observe $ALR^2$ excelling at 8.4 and 7.9 EM gains on HotpotQA and SQuAD, respectively. Furthermore, we conduct extensive analysis revealing that $ALR^2$ displays favourable generalization capability and significantly mitigates retrieval issues, e.g. hallucinations and low accuracy, that are prevalent in prompting-based approaches (§5).

Our contributions are three-fold:

- In our preliminary study, we show that as the input context length grows, LLM reasoning performance degradation is higher than that of retrieval.
- We propose $ALR^2$, a retrieve-then-reason approach to alleviate the challenges occurred when LLMs reasoning over long-contexts by aligning LLMs with both retrieval and reasoning objectives. $ALR^2$ substantially mitigates the primary difficulties of long-context reasoning, such as hallucination and low retrieval accuracy.
- We conduct extensive experiments to systematically compare our approach with baseline methods, including both prompting-based and fine-tuning-based techniques. Our results show that $ALR^2$ significantly outperforms these baselines on long-context QA tasks, i.e., achieving at least least 8.4 and 7.9 EM gains on long-context version of the HotpotQA and SQuAD benchmarks, respectively.

## 2 PRELIMINARY STUDY

In this section, we will present the challenges of long-context reasoning and the motivation for our work, drawn from observations on a series of long-context tasks.

**Setup** We assess the long-context capabilities of LLMs using a series of increasingly challenging tasks, inspired by prior work (Chen et al., 2023; Hsieh et al., 2024):

- **Task 1: Passkey Retrieval**: A sentence, such as "The passkey of Alice is KEY," is inserted into the long context, and the LLM is asked to retrieve Alice's passkey. The "KEY" is a randomly generated string of 10 characters, sampled from the alphabet (A–Z).

- **Task 2: Multi-hop Passkey Retrieval**: In this task, the 10-character passkey is split into two halves, with each half randomly assigned to "Alice" and "Bob." The LLM is then prompted to retrieve either Alice's or Bob's passkey. This task requires an LLM to identify the most relevant one from multiple passkeys, making it more complex than the first.
- **Task 3: Multi-hop Passkey Retrieval + Reasoning**: This task involves both retrieval and reasoning, and Task 1 & 2 are the pre-requisite tasks, i.e., the passkey retrieval is critical for the final reasoning. Given the passkeys of both Alice and Bob, the LLM is asked to concatenate them, e.g., if "Alice has ABC" and "Bob has DEF", then the expected answer is "ABCDEF." The order of concatenation is critical, and the final concatenated answer is controlled to match the passkeys used in Task 1. Notably, the reasoning is designed to be simple purposefully, and a model is expected to achieve good performance on this task if it performs well on Task 1 & 2.

For all tasks, the LLMs are prompted to generate the correct passkey from the long context. We generate 100 random passkeys and use the Wikitext-103 dataset (Grave et al., 2017) to augment the input context. The length of the Wikitext-103 context is varied between 0 and 128K tokens, with the passkeys randomly inserted at 30% and 60% positions within the context. Exact match (EM) is used to evaluate the LLMs' performance. We test three modern LLMs in this experiment: COMMAND-R-PLUS[2], GPT-4 (Achiam et al., 2023), and CLAUDE-3-HAIKU (Anthropic).

**Results** From Figure 1, we make several key observations. First, most LLMs perform well on the first two passkey retrieval tasks (Tasks 1 & 2), achieving high EM scores across context lengths ranging from 0K to 128K. Second, while the simple reasoning required by Task 3 is not an issue for most LLMs within short context, their performance drops significantly as the context length increases. This decline is more pronounced in Task 3 compared to the retrieval-only tasks in Task 1 and 2. Results of Task 3 reveal the challenge facing by modern LLMs in directly reasoning over the long-context input[3]. It further drives our motivation in leveraging an intermediate retrieval step as a stepping stone for the subsequent reasoning step to address the challenge.

## 3 METHODS

### 3.1 OVERVIEW

To address the aforementioned challenges, we introduce a retrieve-then-reason approach which augments the long-context reasoning capability of LLMs via a two-stage breakdown, i.e., (i) first explicitly retrieving relevant facts pertaining to the task; and (ii) then reasoning over the collected facts to derive the final answer. Moreover, as manifested in Figure 2, modern LLMs struggled to retrieve with high accuracy in realistic scenarios and frequently hallucinate *"retrieved facts"*, leading to faulty reasoning and the generation of incorrect answers. To this end, we introduce a simple and effective method, namely ALR$^2$, to augment the long-context reasoning capability of LLMs by an explicit two-stage procedure, i.e., ALigning LLMs with the objectives of both Retrieval and Reasoning. Our approach is inspired by the formulation of Retrieval-Augmented Generation (RAG) approach (Lewis et al., 2020), which was originally designed for open-domain QA.

### 3.2 FORMULATION OF RAG

The standard open-domain QA problem can be formalized as:

$$\boldsymbol{y}^* = \arg\max_{\boldsymbol{y}} p_\theta(\boldsymbol{y}|\boldsymbol{q}),$$

where $\boldsymbol{q}$ is the user question, $\boldsymbol{y}^*$ is the answer with the highest probability, and $\theta$ denotes parameters of the whole system. RAG introduces a latent variable $\boldsymbol{z}$ and reformulates the conditional probability $p_\theta(\boldsymbol{y}|\boldsymbol{q})$ as follows:

$$p_\theta(\boldsymbol{y}|\boldsymbol{q}) = \sum_{\boldsymbol{z}} p_\phi(\boldsymbol{z}|\boldsymbol{q}) p_\mu(\boldsymbol{y}|\boldsymbol{z}, \boldsymbol{q}), \tag{1}$$

---

[2]https://cohere.com/blog/command-r-plus-microsoft-azure

[3]Similar trends are observed in more complex, realistic question-answering tasks, such as HotpotQA and SQuAD (see §4 for more details).

---

**Question**:
Aside from the Apple Remote, what other device can control the program Apple Remote was originally designed to interact with?

**Golden Supporting Facts**:
[1] The Apple Remote is a remote control device released in or after October 2005 by Apple Inc. for use with a number of its products which use infrared capabilities.
[2] The device was originally designed to interact with the Front Row media program on the iSight iMac G5 and is compatible with some later desktop and portable Macintosh computers.
[3] The software relies on iTunes and iPhone and is controlled by an Apple Remote or the *keyboard function keys*.

**Golden Answer**: keyboard function keys

---

**Retrieved Supporting Facts**:

[1] The Apple Remote was originally designed to interact with the Front Row media program.
[2] Additionally, the *Siri Remote*, released in tandem with the fourth generation Apple TV, is also compatible with the program.

**Model Answer**: Siri Remote

---

Figure 2: Error case of the retrieve-then-reason prompting approach. The Command-R model and prompt in Figure 3 are used, and more details are in §4.1. The text with under-wave is the information matched between the golden facts and retrieved facts. We also use the text with underline to represent information that is not in golden facts but in the long context. The information hallucinated by LLM are marked by red text. The *italic text* in supporting facts shows the appearance of the answer.

where $z$ represents the retrieved passage, while $p_\phi(z|q)$ and $p_\mu(y|z,q)$ are the probability distributions of the retriever and generator, respectively. The retriever, parameterized by $\phi$, builds an external retrieval index and retrieves the top-$K$ most relevant passages for each question from the index (Karpukhin et al., 2020; Izacard et al., 2022; Ni et al., 2022). The generator, typically a language model (Vaswani, 2017) parameterized by $\mu$, then takes the retrieved passages and the question $q$ as input to generate the final answer.

## 3.3 ADAPTING RAG FORMULATION TO LONG-CONTEXT REASONING

Given that LLMs perform well in explicit long-context retrieval, a natural solution is to adapt the RAG formulation (Eq. 2) to re-organize the sparsely distributed relevant facts in intermediate steps, and then reason over those retrieved facts. We define our approach for long-context reasoning as follows:

$$p_\theta(y|q,c) = \sum_z p_\theta(z|q,c)p_\theta(y|z,q,c), \qquad (2)$$

where $c$ represents the long context, and both the retriever and the generator are parameterized by $\theta$. Since the key information $z$ is essential for answering the question, the term $p_\theta(z|q,c)$ models the process of explicitly retrieving $z$ from the long context $c$. Once the key information is retrieved, the generator $p_\theta(y|z,q,c)$ can utilize it to answer the question. By leveraging an intermediate retrieval step, our approach simplifies the long-context reasoning to its short-context counterpart, i.e, the reasoning procedure operates on the condensed evidence collected from the retrieval step.[4]

## 3.4 ALIGNING GENERATION TO RETRIEVAL & REASONING

Though Differentiable Search Index (DSI) (Tay et al., 2022; Bevilacqua et al., 2022) demonstrates that replacing the retrieval process with generation is feasible, pre-trained LLMs are generally not

---

[4]In Figure 3, we provide a concrete prompt to implement our approach. As shown in Appendix A, it can already significantly alleviate the performance drop on the reasoning task introduced in preliminary study (§2).

well-aligned with the retrieval objective. As illustrated in Figure 2, simply prompting the LLM to retrieve key information often results in low retrieval accuracy and severe hallucination[5], which in turn leads to flawed reasoning and the production of incorrect answers.

To address these challenges, we align the LLM with both retrieval and reasoning objectives as defined in Eq. 2. To build the training data, we collect datasets that provide the necessary supporting facts (a list of statements crucial for answering the question) and use these facts as the target $z^*$. We then jointly optimize the LLM for both long-context retrieval and answering with the following objective:

$$\mathcal{L}_{align} = \frac{1}{N} \sum_{i=1}^{N} \big( \log p_\theta(z_i^* | q_i, c_i) + \log p_\theta(y^* | z_i^*, q_i, c_i) \big), \tag{3}$$

where $N$ is the number of training examples, and $z_i^*$ and $y_i^*$ are the golden supporting facts and answers for the $i$-th example, respectively. Notably, for questions with multiple supporting facts, such as in multi-hop QA datasets (Yang et al., 2018), the LLM is trained to retrieve all supporting facts by formatting $z^*$ as a concatenated string of bulleted facts. We adopt the prompting template in Figure 3 for our model training.

## 3.5 DISCUSSION

Our retrieval approach differs from others in two key aspects. First, most dense retrieval methods (Karpukhin et al., 2020), which use collections of dense vectors as the index, or Differentiable Search Index (DSI), which leverages model parameters (Tay et al., 2022) for indexing. In contrast, we treat the long input context itself as the retrieval index which supports the update of index on-the-fly. Second, most prior methods do not account for the relationships between retrieved statements and the correct number of statements to retrieve, often relying on large top-$K$ values to ensure all necessary information are retrieved. Differently, our retriever is explicitly trained to retrieve a coherent set of facts that are relevant for answering the question, which is particularly valuable in tasks like multi-hop QA. A better retrieval quality is not only important for the subsequent reasoning, but also provides a more trustworthy rationale, helping users better understand the decision-making process of LLMs.

## 4 EXPERIMENTS

### 4.1 SETUP

We follow the RULER benchmark (Hsieh et al., 2024) to evaluate the long-context performance of our approach and the baselines. Our focus is on question-answering (QA) tasks, but we also assess performance on retrieval tasks, such as variants of the Needle-in-a-Haystack (NIAH) task:

- **QA**: RULER utilizes two QA datasets, i.e., SQuAD (Rajpurkar et al., 2016) and HotpotQA (Yang et al., 2018), to build long-context evaluation datasets, which test the single-hop and multi-hop QA capabilities of LLMs in the long-context scenario.
- **NIAH**: The NIAH task is a synthetic retrieval task commonly used for long-context evaluation. The simplest version involves extracting a "needle", such as an 8-digit number, from a long context. RULER includes four variants of the NIAH task: Single NIAH, Multi-keys NIAH, Multi-values NIAH, and Multi-queries NIAH.

More details about those datasets are shown in Appendix B.1.

We prepared the training data for each task accordingly. For the QA tasks, we used the supporting facts provided by HotpotQA and the sentences containing the answers in SQuAD as the golden retrieval results. Notably, each question in the SQuAD dataset only has one retrieved sentence, while HotpotQA questions typically have 2 to 6 supporting facts. These supporting facts are organized into a string of bulleted items. For the NIAH tasks, we treated the needle sentences as the retrieved results. The number of training examples from HotpotQA, SQuAD, and the four NIAH variants is 5,000, 25,000, and 1,600, respectively. The context lengths (in tokens) of the training examples are 4K, 8K, 16K, and 32K, with the data ratios for each being 1/10, 2/10, 3/10, and 4/10, respectively. Inference for longer contexts is done by extrapolation. More training details are in Appendix B.2. We used

---

[5]See §5.1 for more quantitative analysis for the retrieval.

| Model | Prompt | 4K | 8K | 16K | 32K | 64K | 128K | Overall |
|---|---|---|---|---|---|---|---|---|
| | | | | *HotpotQA* | | | | |
| GPT-3.5 | DA | 52.4 | 49.4 | 46.5 | – | – | – | – |
| GPT-4 | DA | 59.8 | 58.1 | 57.8 | 53.8 | 51.0 | 48.4 | 54.8 |
| GEMINI | DA | 59.4 | 57.3 | 58.72 | 54.5 | 56.7 | 52.6 | 56.5 |
| CLAUDE-3-HAIKU | DA | 49.2 | 48.0 | 42.4 | 43.0 | 38.8 | 49.4 | 45.1 |
| | DA | 51.6 | 51.2 | 47.4 | 47.4 | 43.6 | 34.29 | 45.9 |
| CMD-R | RR | 58.8 | 55.6 | 54.8 | 54.6 | 51.4 | 44.8 | 53.3 |
| | QF | 53.4 | 54.4 | 49.4 | 48.6 | 44.8 | 32.8 | 46.0 |
| | S2A | 43.0 | 44.6 | 42.8 | 40.8 | 34.0 | 30.5 | 39.2 |
| CMD-R-FT | DA | 72.8 | 71.39 | 72.0 | 69.1 | 67.4 | 57.4 | 68.3 |
| ALR$^2$ | RR | **77.2** | **76.8** | **77.2** | **76.6** | **77.5** | **75.2** | **76.7** |
| | | | | *SQuAD* | | | | |
| GPT-3.5 | DA | 63.8 | 60.0 | 53.1 | — | — | — | — |
| GPT-4 | DA | 82.3 | 75.6 | 72.3 | 70.1 | 63.0 | 59.4 | 70.4 |
| GEMINI | DA | 65.9 | 67.2 | 66.5 | 65.6 | 64.6 | 62.5 | 65.4 |
| CLAUDE-3-HAIKU | DA | 67.0 | 63.6 | 60.6 | 53.2 | 53.8 | 55.0 | 58.8 |
| | DA | 65.6 | 61.4 | 63.8 | 62.4 | 54.0 | 49.6 | 59.4 |
| CMD-R | RR | 58.8 | 55.1 | 52.4 | 51.8 | 51.3 | 49.2 | 53.1 |
| | QF | 63.6 | 61.6 | 60.6 | 57.9 | 46.6 | 43.2 | 55.5 |
| | S2A | 42.0 | 42.4 | 49.4 | 46.6 | 41.1 | 38.3 | 43.3 |
| CMD-R-FT | DA | 80.8 | 54.2 | **68.6** | 78.2 | 50.6 | 53.2 | 64.2 |
| ALR$^2$ | RR | **89.8** | **63.0** | 64.0 | **78.2** | **66.6** | **71.2** | **72.1** |
| | | | | *NIAH* | | | | |
| GPT-3.5 | DA | 99.85 | 98.55 | 93.05 | - | - | - | - |
| GPT-4 | DA | 100.0 | 99.9 | 99.65 | 98.5 | 96.1 | 91.4 | 97.60 |
| CMD-R | DA | 94.6 | 91.9 | 89.29 | 94.37 | 93.84 | 87.99 | 92.00 |
| CMD-R-FT | DA | **100.0** | **100.0** | 99.95 | **99.95** | **99.85** | 98.79 | 99.75 |
| ALR$^2$ | RR | **100.0** | **100.0** | **100.0** | **99.95** | **99.85** | **98.99** | **99.79** |

Table 1: Experiments on question-answering (QA) and needle-in-a-haystack (NIAH) tasks. The best results are highlighted by bold text. The evaluation metric is Extract Match (EM).

Exact Match (EM) as our evaluation metric. Following the setup of RULER, we report the average EM score on 500 test cases for each dataset or sub-task. Importantly, to avoid data contamination, we ensured that both the context and the question-answer pairs for testing were unseen during training.

## 4.2 BASELINES

We employ several prompting techniques for the long-context tasks:

- **DA**: This prompting approach is used for Direct Answering, where the model directly generates an answer without explicit retrieval. The template is illustrated in Figure 4.
- **RR**: This prompting method, shown in Figure 3, is used for the Retrieve-then-Reason framework in our work. It first prompts the model to retrieve the key information from the long context and then complete the reasoning to answer the question.
- **QF**: Similar to the RR, Anthropic[6] and Lee et al. (2024) empirically find that guiding the LLM to generate Quotes and citations First, before providing answers, improves long-context reasoning. We evaluate a variant of this approach, as shown in Figure 5.
- **S2A**: Instead of retrieving information directly from the long context, the System-2 Attention approach (Weston & Sukhbaatar, 2023) focuses on answering questions based on summarized and reorganized information extracted from the long context. Our prompt template for this approach is presented in Figure 6.

For our experiments, we use the Command-R (CMD-R) model[7] with 35 billion parameters as our backbone. We evaluate its performance using the four prompting techniques: DA, RR, QF, and S2A. These prompting-based baselines are referred to as CMD-R + X, where X is the name of

---

[6]https://docs.anthropic.com/claude/docs/advanced-text-analysis
[7]https://cohere.com/blog/command-r

| Model | Prompt | 4K | 8K | 16K | 32K | 64K | 128K | Overall |
|-------|--------|----|----|-----|-----|-----|------|---------|
| | | | | *Hallucination Rate* ↓ | | | | |
| CMD-R | RR | 66.57 | 61.04 | 54.54 | 57.01 | 60.79 | 66.74 | 61.1 |
| $ALR^2$ | RR | **0.26** | **0.17** | **0.17** | **0.17** | **0.36** | **0.61** | **0.29** |
| | | | | *Recall* ↑ | | | | |
| CMD-R | RR | 27.37 | 32.50 | 39.53 | 41.35 | 41.34 | 22.28 | 34.06 |
| $ALR^2$ | RR | **69.47** | **69.39** | **69.64** | **69.47** | **67.78** | **66.99** | **68.79** |

Table 2: Evaluation of retrieval quality on HotpotQA dataset. The hallucination rate is the percentage of retrieved sentences that are not presented in the context. The recall score is the percentage of golden facts that are displayed in retrieved sentences. The best results are highlighted by bold text.

the prompt used. Additionally, to ensure a fair assessment on the performance gains otained from model fine-tuning, we include another CMD-R-FT + DA baseline. This model is built by fine-tuning CMD-R on the same training data, described in §4.1, to directly produce the answer, i.e., without a retrieval step, given the input context and the question. For a broader set of evaluations, we also test the DA prompt on other frontier models, including GPT-3.5[8] (Ouyang et al., 2022), GPT-4 (Achiam et al., 2023), GEMINI (Team et al., 2023), and CLAUDE-3-HAIKU (Anthropic).

### 4.3 RESULTS

As shown in Table 1, $ALR^2$ achieves significantly better performance on the two question-answering (QA) benchmarks. Notably, on the multi-hop QA benchmark, HotpotQA, $ALR^2$ maintains consistent performance from 4K to 128K tokens of context. Additionally, the retrieve-then-reason prompting baseline, i.e., CMD-R+RR, shows a smoother performance curve on HotpotQA compared to the direct-answer baseline, i.e., CMD-R+DA. However, on the single-hop SQuAD benchmark, CMD-R+RR performs worse than CMD-R+DA. We hypothesize that this is due to over-retrieval by CMD-R+RR, as it tends to retrieve excessive information that becomes redundant for single-hop questions. In contrast, after aligning the LLM to handle both single-hop and multi-hop questions, our $ALR^2$ approach achieves superior overall performance.

We also compare different two-stage prompting techniques, namely RR, QF, and S2A. As shown in Table 1, CMD-R+RR outperforms CMD-R+QF on HotpotQA but falls short on SQuAD. We find that this discrepancy is closely related to the number of sentences retrieved by each prompt. Specifically, CMD-R+RR retrieves an average of 7.5 sentences across both QA datasets, while CMD-R+QF retrieves only 2.2 sentences. Since HotpotQA is a multi-hop QA dataset that requires more supporting facts, CMD-R+RR performs better on this dataset. Both CMD-R+QF and CMD-R+RR outperform the system-2 attention baseline, i.e., CMD-R+S2A, in our experiments.

In addition, on the NIAH tasks, all evaluated approaches obtain comparable performances. This is unsurprising given the fact that NIAH tasks only require straightforward information retrieval, which is relatively easy for LLMs to handle. When evaluating long-context performance, we caution researchers against averaging the performance of long-context retrieval and reasoning tasks, such as NIAH and QA, due to the significant performance gap between them.

## 5 ANALYSES

### 5.1 RETRIEVAL QUALITY BY LLM

**Setup.** We evaluate the hallucination rate of our retriever, $p_\theta(z|q, c)$, on the HotpotQA dataset. The dataset construction for the long-context setting follows the same procedure as described in the main experiments (§4). The hallucination rate is defined as the percentage of retrieved sentences that

---

[8]Since thye maximum context length of GPT-3.5 is 16K, we only show the performance of GPT-3.5 within this context length.

| Model | Prompt | 4K | 8K | 16K | 32K | 64K | 128K | Overall |
|---|---|---|---|---|---|---|---|---|
| | | *StrategyQA* | | | | | | |
| CMD-R-FT | DA | 61.57 | 60.26 | 60.69 | 60.69 | 58.95 | 56.33 | 59.74 |
| ALR$^2$ | RR | **72.48** | **71.61** | **70.74** | **70.16** | **71.61** | **72.48** | **71.51** |
| | | *TriviaQA* | | | | | | |
| CMD-R-FT | DA | 74.6 | 74.2 | 73.2 | 70.0 | 69.0 | 69.19 | 71.69 |
| ALR$^2$ | RR | **77.0** | **76.4** | **74.4** | **72.60** | **73.94** | **71.98** | **74.38** |

Table 3: EM scores on StrategtQA and TriviaQA tasks. The best results are highlighted by bold text.

do not match any sentences in the context, calculated as:

$$\text{Hallucination}(\mathcal{Z}, \mathcal{C}) = 1 - \frac{1}{N} \sum_{i=1}^{N} \mathbb{1}_{\{z_i \in c_i\}},$$

where $\mathcal{Z} = \{z_i\}_{i=1}^{N}$ is the collection of all retrieved sentences, $z_i$ is a single retrieved sentence, and $\mathcal{C} = \{c_i\}_{i=1}^{N}$ is the corresponding collection of input context. Additionally, since HotpotQA provides golden supporting facts for each question, we evaluate retrieval quality using the recall score, which measures the fraction of golden facts successfully retrieved by the LLM.

**Results.** As shown in Table 2, by aligning generation with retrieval objective, the hallucination rate of our approach is significantly better than the baseline. Additionally, ALR$^2$ also achieves notably higher retrieval recall scores compared to the prompting-based method, highlighting the importance of generation-retrieval alignment. The reduced hallucination rate and improved retrieval recall are not only crucial for enhancing the generator's accuracy, but they also provide a more trustworthy rationale, helping users better understand the decision-making process of LLMs.

## 5.2 GENERALIZATION TO UNSEEN DATA

**Setup.** We are also interested in whether ALR$^2$, trained on the SQuAD (Rajpurkar et al., 2016) and HotpotQA (Yang et al., 2018) datasets, can generalize to unseen datasets. To explore this, we follow the strategy of RULER (Hsieh et al., 2024) to pre-process two additional QA datasets: StrategyQA (Geva et al., 2021) and TriviaQA (Joshi et al., 2017). We compare the performance of the LLM fine-tuned on the direct-answer template (CMD-R-FT + DA) with our approach (ALR$^2$).

**Results.** As shown in Table 3, ALR$^2$ consistently outperforms CMD-R-FT + DA across context lengths ranging from 4K to 128K. These results indicate that ALR$^2$ provides a more robust and stable framework for solving long-context QA problems, even when applied to unseen datasets.

## 6 RELATED WORK

### 6.1 LONG-CONTEXT LARGE LANGUAGE MODEL

Long-context ability is critical for employing a large language model (LLM) to solve real-world problems, e.g., document analysis (Wang et al., 2024a), tool use (Yao et al., 2023; Schick et al., 2023), etc. Many components of an LLM may affect the long-context performance. For instance, researchers find that the design of the position embedding is important for the extrapolation performance of an LLM (Su et al., 2024a; Press et al., 2022; Chen et al., 2023; Ruoss et al., 2023; Men et al., 2024), i.e., testing the LLM on context length that is larger than the training window size. In addition, the attention distribution and scale of the Transformer model (Chiang & Cholak, 2022; Xiao et al., 2024) may also affect the long-context ability.

To extend the context length of LLM, some works try to compress the long context into shorter text (Jiang et al., 2023; 2024; Xu et al., 2024), limited hidden representations (Chevalier et al., 2023; Ge et al., 2024), or model parameters (Wang et al., 2024b), saving the number of input tokens in the context. Moreover, splitting long context into fixed segments and caching them for re-use is

also widely used for increasing the effective context length of LLMs (Dai et al., 2019). Different from Transformer-XL (Dai et al., 2019) that only leverages local attention, Munkhdalai et al. (2024) introduce an additional global momentum to store the knowledge in all the previous context. Yen et al. (2024) leverages a small bi-directional model (Liu, 2019) to encode the segments independently, and allows the LLM to attend to those segments during generation.

In our work, the target is not to extend the context length. In contrast, we find that the performance of QA will degenerate more significantly with longer context for most popular LLMs, similar to the findings in Liu et al. (2024). Therefore, we propose a new formulation inspired by retrieval-augmented generation to alleviate this common issue of LLMs.

Recently, there are also many benchmarks are proposed to evaluate the long-context performance of LLMs (Hsieh et al., 2024; Wang et al., 2024a; Lee et al., 2024; Zhang et al., 2024; Krishna et al., 2023; Karpinska et al., 2024; Kuratov et al., 2024). Considering our focus is the general QA problems and the construction of long-context QA data is similar in many benchmarks, we use RULER (Hsieh et al., 2024) as our main benchmark.

### 6.2 RETRIEVAL-AUGMENTED GENERATION

Retrieval-Augmented generation (RAG) has been extensively used to solve NLP problems, e.g., open-domain question-answering (Lewis et al., 2020; Asai et al., 2024), language modeling (Guu et al., 2020; Lan et al., 2023; Shi et al., 2024), machine translation (Gu et al., 2018; Cai et al., 2021), etc. For most RAG works, the dual-encoder-based retriever (Karpukhin et al., 2020; Izacard et al., 2022; Ni et al., 2022), which measures the similarity between inputs and retrieval documents by their encoded embeddings, has became the *de facto* approach for retrieval. The retrieved top-$K$ documents will be integrated into the generator, e.g., a language model (Radford et al., 2019), to enhance the generation. The most related work in this research line is Xu et al. (2024), which aims to save the number of input tokens to an LLM by compressing the retrieved $K$ documents. However, our work focuses on enhancing the long-context reasoning of LLMs.

### 6.3 CITATION GENERATION

There exist many works that aim to attribute the prediction of an LLM to the information source that the LLM is based on, i.e., citation generation. Some train the model to learn this ability on a standardized citation format (Menick et al., 2022; Taylor et al., 2022; Khalifa et al., 2024; Ye et al., 2024). While others use the instruction-following ability and post-processing techniques to enable this feature at inference (Gao et al., 2023a; Kamalloo et al., 2023; Gao et al., 2023b; Liu et al., 2023).

Our work aims to precisely retrieve the bag of facts that can support the question-answering in long-context scenario, which is also beneficial for the explainability of the prediction. However, the focus of our work is more about enhancing the long-context QA ability of LLMs, which is orthogonal to works in citation generation.

## 7 CONCLUSION

In this work, we addressed the challenge of long-context reasoning in large language models (LLMs) by introducing ALR$^2$, a retrieve-then-reason approach that aligns LLMs with both retrieval and reasoning objectives. Through extensive experiments on both single-hop and multi-hop QA tasks, we demonstrated that ALR$^2$ significantly outperforms existing baselines, especially in long-context scenarios. Our analysis also highlighted the importance of aligning generation with retrieval to reduce hallucinations and improve retrieval accuracy. Moreover, we showed that ALR$^2$ generalizes well to unseen datasets, offering a robust solution for long-context QA problems.

**Limitations.** We also note some of the limitations of our work. For example, as most works based on the RAG formulation, our method is hard to enhance the summarization tasks, in which all the information in the long context is important for the final prediction. In addition, only considering one granularity, i.e., sentence, in the retrieval may be limited in some scenarios. A better way is to allow the users to choose the retrieval granularity, e.g., phrase, sentence, or passage, in the instruction. We leave the addressing of those limitations in future works.

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

```
Stage-1 Input:
Please retrieve all the sentences in the given documents that are important and relevant to answer the
question.

Question: {QUERY}

The following are given documents.

{CONTEXT}

Please retrieve all the sentences in the given documents that are important and relevant to answer the
question. The question is highlighted again at below.

Question: {QUERY}
Retrieved sentences:
(For each retrieved sentence, please start from the bullet symbol "-")

Stage-1 Output:
{MODEL_RETRIEVED_SENTENCES}

Stage-2 Input:
Please answer the question based on the given retrieved information. Only give me the answer and do
not output any other words.
Question: {QUERY}
Answer:

Stage-2 Output:
{ANSWER}
```

Figure 3: The retrieve-then-reason (RR) prompt for long-context QA. The {CONTEXT} is the placeholder for long context and {QUERY} is for user question. The red parts {MODEL_RETRIEVED_SENTENCES} and {ANSWER} are generated by LLM.

```
Stage-1 Input:
Answer the question based on the given documents. Only give me the answer and do not output any
other words.

The following are given documents.

{CONTEXT}

Answer the question based on the given documents. Only give me the answer and do not output any
other words.

Question: {QUERY}

Stage-1 Output:
{ANSWER}
```

Figure 4: The direct-answering (DA) prompt for long-context QA. The CONTEXT is the placeholder for long context, QUERY is for user question, and the red part {ANSWER} is the answer directly generated by LLM.

## A    ADDITIONAL RESULTS FOR PRELIMINARY STUDY

We also evaluate the effectiveness of the retrieve-then-reason framework on the challenging task in the section of preliminary study (§2). As shown in Table 4, simply using the RR prompt (Figure 3)

| Model | Prompt | 16K | 32K | 64K | 128K | Overall |
|---|---|---|---|---|---|---|
| CMD-R-PLUS | DA | 0.70 | 0.16 | 0.0 | 0.0 | 0.21 |
| CMD-R-PLUS | RR | **0.84** | **0.86** | **0.56** | **0.0** | **0.56** |

Table 4: EM scores on Task 3, i.e., Multi-hop Passkey Retrieval + Reasoning, in our preliminary study (§2). The best results are highlighted by bold text.

for the Command-R-Plus model can improve the overall EM score from 0.21 to 0.56, significantly alleviating the performance drop on the task that requires long-context reasoning.

# B    DETAILS OF EXPERIMENT SETUP

## B.1    RULER TASKS

**QA Tasks**    We use the validation sets of SQuAD (Rajpurkar et al., 2016) and HotpotQA (Yang et al., 2018) datasets to build the long-context benchmarks, where each question in those datasets is associated with some relevant passages. Following Hsieh et al. (2024), We use concatenated and shuffled passages as the context, which include both relevant passages and randomly sampled ones.

**NIAH Tasks**    Following Hsieh et al. (2024), we consider four kinds of Needle-in-a-Haystack (NIAH) tasks in our work and report the averaged performance of them:

- **Single NIAH (S-NIAH)**: The "haystack" text is from Paul Graham essays (Kamradt, 2023) is from "needle" sentence of this task is "One of the special magic numbers for {KEY} is: {VALUE}", where the "{KEY}" and "{VALUE}" are placeholders for English words and 7-digit numbers, respectively. For S-NIAH, we only insert one needle sentence into the long-context.

- **Multi-keys NIAH (MK-NIAH)**: In this setting, we insert four needle sentences into the context and ask an LLM to retrieve only one of them. Each of the four needle sentences has unique "{KEY}" and "{VALUE}".

- **Multi-values NIAH (MV-NIAH)**: In this setting, part of the needle sentences may share the same key. We will ask an LLM to retrieve all the values associated with the same key.

- **Multi-queries NIAH (MQ-NIAH)**: Compared with MK-NIAH, we need the LLM to retrieve multiple values in needle sentences associated with specific keys.

## B.2    TRAINING DETAILS

During training, we set the base value of rotary position embedding to $8e6$ (Men et al., 2024; Su et al., 2024b). The training batch size is 64 on V4-512 TPU. For every 3 updates on the prepared training dataset, we will also update our parameters on pre-training data with 64K tokens for one step. For both of the two-stage and one-stage fine-tuning methods, we will train the command-r model with 35 billion parameters for 3 epochs.

**Stage-1 Input**:
I'm going to give you some documents. Then I'm going to ask you a question about it. I'd like you to first write down exact quotes of parts of the document that would help answer the question, and then I'd like you to answer the question using facts from the quoted content. Here is the document:

<document>
{CONTEXT}
</document>

First, find the quotes from the document that are most relevant to answering the question, and then print them in numbered order. Please start the quotes with "Relevant quotes:".

Then, answer the question, starting with "Answer:". Do not include or reference quoted content verbatim in the answer. Don't say "According to Quote [1]" when answering. After "Answer:", please only give me the answer and do not output any other words.

Thus, the format of your overall response should look like what's shown between the tags. Make sure to follow the formatting and spacing exactly.

<example>
Relevant quotes:
 [1] "Company X reported revenue of $12 million in 2021."
 [2] "Almost 90% of revenue came from widget sales, with gadget sales making up the remaining 10%."

Answer: $12 million [1].
</example>
Here is the first question: {QUERY}

Give me the quotes and answer for the question immediately without preamble.

**Stage-1 Output**:
{QUOTES_AND_ANSWER}

Figure 5: The quotes-and-citation-first (QF) prompt for long-context QA. The {CONTEXT} is the placeholder for long context and {QUERY} is for user question. The format of {QUOTES_AND_ANSWER} is similar to the example embraced by <example> and </example>.

**Stage-1 Input**:
Given the following text by a user, extract the part that is unbiased and not their opinion winthin a single paragraph, so that using that text alone would be good context for providing an unbiased answer to the question portion of the text.

————————- Start of Text by User ————————-
Answer the question based on the given documents. Only give me the answer and do not output any other words.

The following are given documents.

{CONTEXT}

Answer the question based on the given documents. Only give me the answer and do not output any other words.

Question: {QUERY}

————————- End of Text by User ————————-

Given the above text by a user, extract the part that is unbiased and not their opinion winthin a single paragraph, so that using that text alone would be good context for providing an unbiased answer to the question portion of the text.

Please include the actual question or query that the user is asking. Separate this into two categories labeled with "Unbiased text context (includes all content except user's bias):" and "Question/Query (does not include user bias/preference):".

**Stage-1 Output**:
{COMPRESSED_TEXT}

**Stage-2 Input**:
% Previous Chat History Will Be Discarded %
{COMPRESSED_TEXT}

**Stage-2 Output**:
{ANSWER}

Figure 6: The system-2 attention (S2A) prompt for long-context QA. The {CONTEXT} is the placeholder for long context and {QUERY} is for user question. The {COMPRESSED_TEXT} contains the copied question and the compressed long context based on the question. In stage-2, we only input the {COMPRESSED_TEXT} to LLM, discarding the previous chat history.