# OpenReview forum: "ALR$^2$: A Retrieve-then-Reason Framework for Long-context Question Answering"
_ICLR.cc/2025/Conference — Submitted to ICLR 2025_

### Official Review · Reviewer_WCoe · 2024-11-02

**Soundness:** 2
**Presentation:** 3
**Contribution:** 2
**Rating:** 6
**Confidence:** 4

**Summary:**

The paper presents $ALR^2$, a retrieve-and-reason framework for QA over long contexts. $ALR^2$ is trained to first retrieve relevant information from a long context, and then reason over it to arrive at an answer. With Command-R, $ALR^2$ significantly outperforms prompted variants and a model directly trained to generate an answer on HotpotQA and SQuAD (in a variant where gold facts are augmented to create a long context), and generalizes to unseen StrategyQA and TriviaQA datasets.

**Strengths:**

- The paper deals with an important subject, long-range QA.
- The improvements over a variant trained to directly generate the answer are significant, and improvements generalize to unseen datasets.
- The paper is well-written and easy to follow.

**Weaknesses:**

- While the generalization study presented in Section 5.2 is important, it only experiments with datasets that were “modified” to have a long-context, in a method that is similar to those of the training data. I believe the paper could benefit from experimenting with QA datasets that have a long context by design, such as those in the SCROLLS benchmark [1]. Additionally, the paper focuses on reasoning, but experiments with a single multi-hop QA benchmark in each setting (in-domain and out-of-domain).

- The results in Tab.1 only compare against all baselines for the trained Command-R models, and models prompted to directly generate the answer for other models. I believe the paper could benefit from experimenting with another trained LLM with all variants, or at least adding results with the ‘retrieve-then-reason’ prompt with current strong LLMs (e.g., Llama-405B, GPT4-o, Claude-Sonnet-3.5).

- Perhaps I am missing something, but as far as I am aware the idea that a LLM is trained to first retrieve information from a long context and then reason over it has been presented previously, for example by training LLMs as re-rankers (e.g., [2]). While there are differences between these two approaches (in $ALR^2$ the LLM generates relevant sentences instead of ranking), it seems there are also similarities (the LLM first identifies relevant information from a long context). Hence, the paper might benefit from additional experiments or analysis that highlight what leads to the improvements of $ALR^2$ (see questions).

- While the paper is overall well-written, there are a few parts that I think could be further improved. Mainly, the preliminary study provides motivation for the work, but lacks a qualitative analysis, so it was not clear to me why models struggle on Task 3. Additionally, I felt like the discussion in Section 3.5 was a bit misleading, as dense retrievers are often much more efficient than the presented approach.


[1] SCROLLS: Standardized CompaRison Over Long Language Sequences

[2] RankZephyr: Effective and Robust Zero-Shot Listwise Reranking is a Breeze!

**Questions:**

Regarding weakness 1 - do you think there are QA datasets that have long-context by design and might be relevant for evaluation?

Regarding weakness 3 - do you think there are other experiments/ablations that are similar to current methods that can be added? For example, the model can be trained to first re-rank the long context (variants that generate relevant indexes for passages like in RankZephyr or relevant snippets as in the paper can be explored). Additionally, an ablation where the final answer is generated by a short-context QA model can help understand if the full context is necessary for reasoning after the retrieval step, or if only the information in the retrieval step is sufficient to answer the questions.

Additionally, perhaps I missed something, but will the model and data be publicly available?

Also, I believe the paper could benefit from a discussion regarding reproducibility and an ethics statement.

---

> ### Author Response · Authors · 2024-11-22
> **Response to WCoe (1/2)**
>
> ### Q1: While the generalization study presented in Section 5.2 is important, it only experiments with datasets that were “modified” to have a long-context, in a method that is similar to those of the training data. I believe the paper could benefit from experimenting with QA datasets that have a long context by design, such as those in the SCROLLS benchmark [1]. Additionally, the paper focuses on reasoning, but experiments with a single multi-hop QA benchmark in each setting (in-domain and out-of-domain). (Also see the first comment in the Questions section)
>
> Following your suggestion, we further evaluated our Retrieve-then-Reason (RR) method on the NarrativeQA dataset, which has the longest context among the QA datasets in the SCROLLS benchmark. The average number of words in the input context for the NarrativeQA dataset is 51,653.
>
> **Setup**    We randomly sampled 100 examples from the validation set of NarrativeQA. We apply both the Retrieve-then-Reason (RR) and Direct-Answer (DA) methods to Command-R model. Following the setting in [1], we input the full story into the model and use corpus-level multi-reference BLEU-1 and BLEU-4 scores as the evaluation metric.
>
> **Results**    As shown in the following table, the RR approach outperforms the DA approach on both BLEU-1 and BLEU-4, demonstrating its potential in enhancing the long-context reasoning in realistic QA datasets.
>
> |  | BLEU-1 | BLEU-4 |
> | --- | --- | --- |
> | DA | 31.74 | 7.57 |
> | RR | 34.68 | 7.78 |
>
> [1]: The NarrativeQA Reading Comprehension Challenge
>
> ### Q2: The results in Tab.1 only compare against all baselines for the trained Command-R models, and models prompted to directly generate the answer for other models. I believe the paper could benefit from experimenting with another trained LLM with all variants, or at least adding results with the ‘retrieve-then-reason’ prompt with current strong LLMs (e.g., Llama-405B, GPT4-o, Claude-Sonnet-3.5).
>
> Thanks for your suggestion. We further evaluate the Retrieve-then-Reason (RR) method on GPT-4 and Claude-3. As shown in the following table, the RR method achieves consistent performance gain on GPT-4 and Claude-3, compared with the Direct-Answering (DA) method. Note that the experiments on 128K is still under-running because of the time limit.
>
> | Model | Prompt | 4K | 8K | 16K | 32K | 64K |
> | --- | --- | --- | --- | --- | --- | --- |
> | GPT-4 | DA | 59.8 | 58.1 | 57.8 | 53.8 | 51.0 |
> | GPT-4 | RR | 64.0 | 63.4 | 62.8 | 61.8 | 54.9 |
> | Claude-3-Haiku | DA | 49.2 | 48.0 | 42.4 | 43.0 | 38.8 |
> | Claude-3-Haiku | RR | 55.2 | 53.4 | 50.0 | 48.0 | 42.1 |
> | CMD-R | DA | 51.6 | 51.2 | 47.4 | 47.4 | 43.6 |
> | CMD-R | RR | 58.8 | 55.6 | 54.8 | 54.6 | 51.4 |
>
> (Experiments on HotpotQA dataset)
>
> ### Q3: Perhaps I am missing something, but as far as I am aware the idea that a LLM is trained to first retrieve information from a long context and then reason over it has been presented previously, for example by training LLMs as re-rankers (e.g., [2]). While there are differences between these two approaches (in  the LLM generates relevant sentences instead of ranking), it seems there are also similarities (the LLM first identifies relevant information from a long context). Hence, the paper might benefit from additional experiments or analysis that highlight what leads to the improvements of  (see questions).
>
> We agree that [2] is a related work, as the retrieved results could be derived from the ranked results. We will cite this work and discuss the differences in our paper. In summary, unlike [2], which focuses on re-ranking at the passage level, our approach trains the LLM to retrieve precise information, such as key sentences from different passages, to support generation. This is because not all information within a passage is necessarily useful. Employing a re-ranking approach becomes burdensome for fine-grained retrieval, such as at the sentence or span level. Additionally, our approach has the potential to extend retrieval to the span level.
>
> [2] RankZephyr: Effective and Robust Zero-Shot Listwise Reranking is a Breeze!

---

> > ### Author Response · Authors · 2024-11-22
> > **Response to WCoe (2/2)**
> >
> > ### Q3.1 Regarding weakness 3 - do you think there are other experiments/ablations that are similar to current methods that can be added? For example, the model can be trained to first re-rank the long context (variants that generate relevant indexes for passages like in RankZephyr or relevant snippets as in the paper can be explored). Additionally, an ablation where the final answer is generated by a short-context QA model can help understand if the full context is necessary for reasoning after the retrieval step, or if only the information in the retrieval step is sufficient to answer the questions.
> >
> > As you suggested, we add two variants of RR method for the ablation study:
> >
> > 1. **CMD-R with RAG**: Instead of using LLM for retrieval, we also use the external Dense Passage Retriever (DPR) [3] to retrieve top-5 passages from the context, and then let the LLM to answer the question based on the retrieved information. This method is to evaluate the effectiveness of the retrieval by LLM.  It is worth noting that, for a fair comparison, our implementation here only retrieves from the passages in the long context, instead of building a large scale datastore based on additional resources.
> > 2. **RR w/o LC**: Compared with standard RR method, we remove the long context in the first-round user input after the generation of retrieval, namely RR w/o LC. This is to understand whether the full context is necessary for reasoning.
> >
> > As shown in the following table, The CMD-R with RAG approach achieves stable performance across the context lengths. However, its performance has a big gap compared with RR method. In addition, though RR w/o LC achieves performance better than DA consistently, removing the long context would hurt the performance of reasoning when comparing with RR method.
> >
> > | Model | Prompt | 4K | 8K | 16K | 32K | 64K |
> > | --- | --- | --- | --- | --- | --- | --- |
> > | CMD-R | DA | 51.6 | 51.2 | 47.4 | 47.4 | 43.6 |
> > | CMD-R | RR | 58.8 | 55.6 | 54.8 | 54.6 | 51.4 |
> > | CMD-R | RAG | 47.4 | 48.4 | 49.2 | 46.8 | 45.6 |
> > | CMD-R | RR w/o LC | 55.6 | 53.0 | 56.0 | 51.2 | 49.8 |
> > (Experiments on HotpotQA dataset)
> >
> > [3]: Dense Passage Retrieval for Open-Domain Question Answering
> >
> > ### Q4: While the paper is overall well-written, there are a few parts that I think could be further improved. Mainly, the preliminary study provides motivation for the work, but lacks a qualitative analysis, so it was not clear to me why models struggle on Task 3. Additionally, I felt like the discussion in Section 3.5 was a bit misleading, as dense retrievers are often much more efficient than the presented approach.
> >
> > There are several kinds of errors in Task 3 (multi-hop passkey retrieval + reasoning):
> >
> > Type 1: wrong order of the concatenated passkey, e.g., we expect ABCDEF but the model generates DEFABC:
> >
> > ```python
> > {
> >     "response": "QFPGWVHITG",
> >     "label": "VHITGQFPGW",
> > },
> > ```
> >
> > Type 2: partial generation, e.g., we expect ABCDEF but the model only partially generates DEF:
> >
> > ```python
> > {
> >     "response": "NWJAU",
> >     "label": "YCMCANWJAU",
> > },
> > ```
> >
> > Type 3: hallucination, e.g., generating nonsense response
> > ```
> > {
> >     "response": "GHKTSAK",
> >     "label": "YCMCANWJAU",
> > },
> > ```
> >
> > Type 4: instruction-following error, e.g., the model directly copy our example in the illustration as the response.
> > ```
> > {
> >     "response": "ABCDEF",
> >     "label": "OVYEAGPEPP",
> > },
> > ```
> >
> > Regarding your concern about the LLM-based retrieval, we add the baseline of RAG method again for a more comprehensive comparison. Though RAG approach is more efficient, we hypothesize that LLM has a better measurement about the relevance between question and context information:
> > | Model | Prompt | 4K | 8K | 16K | 32K | 64K |
> > | --- | --- | --- | --- | --- | --- | --- |
> > | CMD-R | RR | 58.8 | 55.6 | 54.8 | 54.6 | 51.4 |
> > | CMD-R | RAG | 47.4 | 48.4 | 49.2 | 46.8 | 45.6 |

---

> ### Comment · Reviewer_WCoe · 2024-11-25
>
> Thank you for the detailed response! It indeed addresses most of my concerns.
>
> My main remaining concern regards the lack of a re-ranking baseline. If I understand correctly, the new RAG baseline is relatively weak (DPR only and performs worse than directly answering the question when context size is smaller than 32K), while stronger baselines exist (e.g., re-ranking). I agree with the authors that the presented method is more fine-grained and could have additional benefits, but I believe the paper could benefit from empirically examining the differences.

---

> > ### Author Response · Authors · 2024-11-27
> > **Response to Reviewer WCoe (Round 2)**
> >
> > We followed your suggestion and evaluated a stronger RAG baseline by using the retriever from RankZephyr [1] as the retriever.
> >
> > **Setup**: We use the same prompt in RankZephyr [1] to rerank the passages:
> >
> > ```python
> > RERANK_PROMPT = """You are RankLLM, an intelligent assistant that can rank passages based on their relevancy to the query.
> >
> > I will provide you with {num} passages, each indicated by a numerical identifier []. Rank the passages based on their relevance to
> > the search query: {query}
> >
> > {passages}
> >
> > Search Query: {query}
> >
> > Rank the {num} passages above based on their relevance to the search query. All the passages should be included and listed
> > using identifiers, in descending order of relevance. The output format should be [] > [], e.g., [4] > [2]. Only respond with the ranking results, do not say any word or explain."""
> > ```
> >
> > Regarding the representation of each document, we also convert our format `Document N: … {passage} …` to `[N]: … {passage} …`, where N is the id of the passage. We use the top-5 documents introduced from the rank list as the retrieval results and integrate them into the QA prompt.
> >
> > **Results**: Experiments on HotpotQA using CMD-R model as the backbone
> >
> > |  | 4K | 8K | 16K | 32K | 64K |
> > | --- | --- | --- | --- | --- | --- |
> > | DA | 51.6 | 51.2 | 47.4 | 47.4 | 43.6 |
> > | RR | 58.8 | 55.6 | 54.8 | 54.6 | 51.4 |
> > | RR /wo LC | 55.6 | 53.0 | 56.0 | 51.2 | 49.8 |
> > | DPR-RAG | 47.4 | 48.4 | 49.2 | 46.8 | 45.6 |
> > | RankZephyr-RAG | 51.9 | 51.2 | 50.5 | 46.8 | 43.4 |
> >
> > We find that RankZephyr-RAG achieves better performance than DPR-RAG approach, as reviewer WCoe expected. However, both the RR and RR /wo LC methods outperform RankZephyr-RAG method, indicating that the generative retrieval achieves more stable performance in the long-context scenario.
> >
> > We hope the updated information could help you reevaluate our work. We really enjoy exploring the professional suggestions from you. If you have additional questions, please feel free to let us know.
> >
> > [1]: RankZephyr: Effective and Robust Zero-Shot Listwise Reranking is a Breeze!

---

> ### Comment · Reviewer_WCoe · 2024-11-29
>
> Thank you for adding this experiment.
>
> I am a bit surprised that DPR-RAG performs better or similarly than RankZephyr-RAG, as this seems to contradict the original paper (and other works showing that re-ranking with stronger models outperform dense retrievers).
>
> To summarize, I believe the paper could benefit from (a) adding stronger baselines that are commonly used in previous works, (b) discussing these works and how they differ from $ALR^2$, and (c) focusing on comparison to stronger retriever-rerank baselines in addition to direct-answering. Nevertheless, the proposed method has merits and I appreciate the authors' efforts to improve the paper during the discussion period. I updated my score to reflect these changes.

---

> > ### Author Response · Authors · 2024-11-29
> > **Thanks for your kind feedback**
> >
> > Many thanks for your kind feedback! We will revise the paper according to your comments.
> >
> > Moreover, the results of those RAG-based baselines are also interested to us. We hypothesize that this is because of the setting shift from open-domain tasks to long-context tasks.  The standard RAG method is to retrieve additional information from external datastore, i.e., **adding information to the original LLM**. Thus, the LLM could easily benefit from the information that is not shown in the original input. However, in the long-context setting, the RAG model is actually removing redundant passages, i.e., **subtracting information from the original LLM**. Therefore, outperforming the DA prompt, which has the full context, is not easy, especially in short-context scenarios.
> >
> > Thank you again, and we really enjoy the discussion and sincerely appreciate your professional suggestions for our work.

---

### Official Review · Reviewer_Q28w · 2024-11-03

**Soundness:** 2
**Presentation:** 3
**Contribution:** 2
**Rating:** 5
**Confidence:** 4

**Summary:**

This paper proposes a retrieve-then-reason method to address the issue that LLMs (Large Language Models) struggle to handle long contexts properly, especially in reasoning. This proposed approach improves the handling of long contexts by explicitly retrieving supporting facts from the given long context as well as by aligning LLMs with retrieval and reasoning objectives at the same time.

**Strengths:**

- The paper is generally well-written, easy to understand, and clearly presented.
- The preliminary study provides interesting insights into where issues arise when LLMs handle long contexts, showing that while retrieval itself is not problematic, the issues occur during reasoning.

**Weaknesses:**

In my opinion, there seem to be some areas for improvement in the design of the experiments aimed at proving the claims of this paper. If these below points can be clarified through additional responses from the authors, I would be glad to adjust my evaluation accordingly.

1. Details about the experiments are missing. For example, there is insufficient information regarding the hyperparameters used during model training (e.g., learning rate, seed) and the specific training data. It is unclear whether ALR was trained separately on each dataset (e.g., SQuAD, HotpotQA) or trained using both datasets together (according to line 407). If it was the former, which model was used in the experiment described in section 5.2?

2. The experiment setting appears to be incomplete. It seems feasible to apply the RR method (e.g., using multi-turn conversation) to the frontier models utilized in the study (GPT-3.5, GPT-4, GEMINI, CLAUDE-3). Additionally, the benefits of aligning with both objectives are not clearly demonstrated. Why should the LLM be aligned to both objectives? It seems plausible to fine-tune the LLM separately for each objective and then combine the results (e.g., passing the output of the retrieval LLM to the reasoning LLM). An ablation study addressing this approach should be provided.

3. There may not have been a fair comparison with the baselines. It appears that there is a difference in the answer template between the DA prompt (figure 4) and the RR prompt (figure 3), as the "Answer:" is missing in figure 3. Given that LLMs can be sensitive to prompts, this could lead to performance differences. To provide evidence that the comparisons in the experiment were fair, results using the same answer template should be included. Additionally, based on figure 3 and equation (2), it seems that the given long context was used alongside the retrieved facts. A comparison showing the performance when the given long context (i.e., prior chat history) is removed would be necessary for a fair comparison with S2A, which also omits prior records.

**Questions:**

- Lines 357-359 point out that QF only retrieves 2.2 sentences on average (compared to 7.5 for RR). However, in HotpotQA, it is common for only a small number of supporting facts (1-2) to be necessary [1] (even if 2-6 supporting facts are provided). To assess whether the retrieval was truly effective, could the performance of CMD-R QF also be included in the experiment results of Section 5.1? It is particularly important to include not only the recall score but also precision. Relying solely on the recall score could bias the evaluation toward retrieval methods that select a larger number of facts.

[1] [MuSiQue: Multihop Questions via Single-hop Question Composition] (Trivedi et al., TACL 2022)

---

> ### Author Response · Authors · 2024-11-22
> **Response to Reviewer Q28w (1/2)**
>
> ### Q1: Details about the experiments are missing. For example, there is insufficient information regarding the hyperparameters used during model training (e.g., learning rate, seed) and the specific training data. It is unclear whether ALR was trained separately on each dataset (e.g., SQuAD, HotpotQA) or trained using both datasets together (according to line 407). If it was the former, which model was used in the experiment described in section 5.2?
>
> Thanks for pointing out. We will add the following training details in the revised version:
> | Hyper-Parameter | Value |
> | --- | --- |
> | Learning Rate | 1e-5 |
> | LR Schedule | Cosine |
> | Optimizer | Adam |
> | Adam Betas | (0.9, 0.95) |
> | Adam Eps | 1e-8 |
> | Weight Decay | 0.1 |
> | Gradient Clip Norm | 1.0 |
>
> Please feel free to let us know if there are any additional hyper-parameters you want to know.
>
> Moreover, our ALR$^2$ is trained on the mixed dataset, i.e., training one model for all the tasks. Therefore, the model used in Sec 5.2 is the same as the one used in the main experiments.
>
> ### Q2: The experiment setting appears to be incomplete. It seems feasible to apply the RR method (e.g., using multi-turn conversation) to the frontier models utilized in the study (GPT-3.5, GPT-4, GEMINI, CLAUDE-3). Additionally, the benefits of aligning with both objectives are not clearly demonstrated. Why should the LLM be aligned to both objectives? It seems plausible to fine-tune the LLM separately for each objective and then combine the results (e.g., passing the output of the retrieval LLM to the reasoning LLM). An ablation study addressing this approach should be provided.
>
> Following your suggestion, we further evaluate the Retrieve-then-Reason (RR) method on GPT-4 and Claude-3. As shown in the following table, the RR method achieves consistent performance gain on GPT-4 and Claude-3, compared with the Direct-Answering (DA) method. Note that the experiments on 128K is still under-running because of the time limit.
>
> | Model | Prompt | 4K | 8K | 16K | 32K | 64K |
> | --- | --- | --- | --- | --- | --- | --- |
> | GPT-4 | DA | 59.8 | 58.1 | 57.8 | 53.8 | 51.0 |
> | GPT-4 | RR | 64.0 | 63.4 | 62.8 | 61.8 | 54.9 |
> | Claude-3-Haiku | DA | 49.2 | 48.0 | 42.4 | 43.0 | 38.8 |
> | Claude-3-Haiku | RR | 55.2 | 53.4 | 50.0 | 48.0 | 42.1 |
> | CMD-R | DA | 51.6 | 51.2 | 47.4 | 47.4 | 43.6 |
> | CMD-R | RR | 58.8 | 55.6 | 54.8 | 54.6 | 51.4 |
>
> (Experiments on HotpotQA dataset)
>
> We agree that it is indeed feasible to train two separate models for retrieval and reasoning, respectively. This approach aligns with the implementation of RAG methods, which typically consist of a dual-encoder-based retriever and a language model. However, enabling a single general-purpose LLM to handle both retrieval and reasoning within a unified framework, namely auto-regressive generation, offers significant advantages in terms of efficiency and practicality in real-world scenarios. This eliminates the need to deploy and manage two LLM services simultaneously. We will clarify this point further in the revised paper.

---

> > ### Author Response · Authors · 2024-11-22
> > **Response to Reviewer Q28w (2/2)**
> >
> > ### Q3: There may not have been a fair comparison with the baselines. It appears that there is a difference in the answer template between the DA prompt (figure 4) and the RR prompt (figure 3), as the "Answer:" is missing in figure 3. Given that LLMs can be sensitive to prompts, this could lead to performance differences. To provide evidence that the comparisons in the experiment were fair, results using the same answer template should be included. Additionally, based on figure 3 and equation (2), it seems that the given long context was used alongside the retrieved facts. A comparison showing the performance when the given long context (i.e., prior chat history) is removed would be necessary for a fair comparison with S2A, which also omits prior records.
> >
> > Thanks for the good catch. This is actually a typo. In the real DA prompt, we also have the “Answer” which is missed in the illustration figure.
> >
> > As you suggested, we add a variant of RR method that removes the long context in the first-round user input after the retrieval, namely RR w/o LC. As shown in the following table, removing the long context would hurt the performance of RR method slightly. However, the RR w/o LC outperforms the S2A method consistently by a large margin.
> >
> > | Model | Prompt | 4K | 8K | 16K | 32K | 64K |
> > | --- | --- | --- | --- | --- | --- | --- |
> > | CMD-R | DA | 51.6 | 51.2 | 47.4 | 47.4 | 43.6 |
> > | CMD-R | RR | 58.8 | 55.6 | 54.8 | 54.6 | 51.4 |
> > | CMD-R | RR w/o LC | 55.6 | 53.0 | 56.0 | 51.2 | 49.8 |
> > | CMD-R | S2A | 43.0 | 44.6 | 42.8 | 40.8 | 34.0 |
> > (Experiments on HotpotQA dataset)
> >
> > ### Q4: Lines 357-359 point out that QF only retrieves 2.2 sentences on average (compared to 7.5 for RR). However, in HotpotQA, it is common for only a small number of supporting facts (1-2) to be necessary [1] (even if 2-6 supporting facts are provided). To assess whether the retrieval was truly effective, could the performance of CMD-R QF also be included in the experiment results of Section 5.1? It is particularly important to include not only the recall score but also precision. Relying solely on the recall score could bias the evaluation toward retrieval methods that select a larger number of facts.
> >
> > It is true that only a small number of supporting facts may be critical in HotpotQA. However, retrieving just a few sentences may fail to capture all the essential supporting facts, thereby impacting the final performance. Including the precision score, however, poses challenges because the LLM may generate retrieval results accompanied by model explanations. Consequently, the precision score is likely to be underestimated compared to recall. Since recalling critical facts is more crucial for the reasoning phase, we rely on recall for evaluation in this work.

---

> > > ### Comment · Reviewer_Q28w · 2024-11-27
> > >
> > > Thank you for your response. While some of my concerns have been addressed, I still feel that the response to Q4 is insufficient.
> > >
> > > I understand that time is limited, but as I initially asked, could CMD-R QF be added to Table 2?
> > >
> > > Also, due to the nature of HotpotQA, a small number of supporting facts may be sufficient. To demonstrate this, it is necessary to consider not only recall but also the precision scores of sentences retrieved by QF and RR.
> > >
> > > While I understand the authors' argument, I would like to mention that I am not suggesting that precision scores are the only important factor; rather, I believe that both recall and precision should be considered comprehensively. Relying on a single metric alone could bias the evaluation toward retrieval methods that select a larger number of facts.

---

> ### Author Response · Authors · 2024-11-27
> **Response to Reviewer Q28w (Round 2)**
>
> > I understand that time is limited, but as I initially asked, could CMD-R QF be added to Table 2?
>
> Yes, we will add CMD-R QF to Table 2.
>
> -----------
> Following your suggestion, we conducted the analysis in Table 2 for QF method, and also add the ngram-level precision metric, i.e., the percentage of 4-grams in the retrieved sentences that also appear in the golden-truth supporting facts. Below are the results:
>
>
> **Recall**
> | Prompt | 4K | 8K | 16K | 32K | 64K | 128K | AVG. |
> | --- | --- | --- | --- | --- | --- | --- | --- |
> | QF | 18.11 | 19.85 | 22.00 | 21.00 | 19.02 | 11.68 | 18.61 |
> | RR | 27.37 | 32.50 | 39.53 | 41.35 | 41.34 | 22.28 | 34.06 |
>
>
> **Precision (4-gram)**
> | Prompt | 4K | 8K | 16K | 32K | 64K | 128K | AVG. |
> | --- | --- | --- | --- | --- | --- | --- | --- |
> | QF | 56.08 | 57.22 | 53.73 | 50.16 | 46.50 | 30.53 | 49.03 |
> | RR | 41.35 | 42.62 | 41.88 | 40.05 | 34.07 | 33.54 | 38.91 |
>
> We found that RR performs better in terms of recall, while QF excels in precision. This supports our hypothesis that RR is more effective for multi-hop QA, whereas QF is better suited for single-hop QA.
>
> We really appreciate your suggestions to our work, and will integrate those results into the revised version. We hope our results could address your concerns. If you have additional questions, please feel free to let us know.

---

> > ### Comment · Reviewer_Q28w · 2024-11-27
> >
> > Thank you for your response. However, I find it a bit confusing, so I would appreciate it if you could clarify. Is there a specific reason why you used an n-gram-level precision metric? Did you also use an n-gram-level metric for recall?

---

> > > ### Author Response · Authors · 2024-11-27
> > > **Clarify for the n-gram-level precision score**
> > >
> > > Thanks for your prompt reply.
> > >
> > > > Did you also use an n-gram-level metric for recall?
> > >
> > > No, the evaluation of recall is at the sentence level.
> > >
> > > > Is there a specific reason why you used an n-gram-level precision metric?
> > >
> > > We are sorry for the confusion. In our first-round reply, we talked about the challenge when evaluating precision `Including the precision score, however, poses challenges because the LLM may generate retrieval results accompanied by model explanations.` We thought this behavior of LLM may under-estimate the precision score. Therefore, we thought about measuring the precision at the n-gram level.
> > >
> > > If you prefer the sentence-level precision, below are the results:
> > >
> > > Precision (Standard)
> > > | Prompt | 4K    | 8K    | 16K   | 32K   | 64K   | 128K  | AVG.  |
> > > |--------|-------|-------|-------|-------|-------|-------|-------|
> > > | QF     | 37.26 | 37.82 | 35.49 | 35.89 | 30.70 | 17.73 | 32.48 |
> > > | RR     | 18.95 | 22.20 | 24.11 | 23.96 | 21.44 | 16.53 | 21.19 |

---

> > > > ### Comment · Reviewer_Q28w · 2024-11-27
> > > >
> > > > Thank you for the clarification. In my opinion, for multi-hop tasks like HotpotQA, which primarily involve 2-hop reasoning, a higher precision in QF could be advantageous. It would be great to see further analysis, such as the Hallucination Rate in Table 2.
> > > >
> > > > Regardless, I appreciate your active engagement in addressing my questions and concerns, and I have adjusted my score accordingly.

---

> > > > > ### Author Response · Authors · 2024-11-27
> > > > > **thanks for adjusting the score**
> > > > >
> > > > > Thanks for adjusting the score! And we will also add the results for hallucination.
> > > > >
> > > > >
> > > > > As a retriever, higher precision is likely more advantageous. However, the primary goal of this work is to enhance long-context reasoning, where retrieval is a sub-task for the gathering of key information. Therefore, better recall may be more suitable for further reasoning, as it ensures that no key information is neglected. We will also clarify this motivation further.
> > > > >
> > > > > Again, thanks for the suggestion of evaluating other prompts, which is helpful for better understanding the effect of different prompt.

---

### Official Review · Reviewer_4gL5 · 2024-11-05

**Soundness:** 3
**Presentation:** 3
**Contribution:** 3
**Rating:** 6
**Confidence:** 4

**Summary:**

This paper focus on addressing the challenge of LLMs' degrading performance when reasoning over long contexts. This paper proposes a two-stage retrieve-then-reason approach: first retrieving relevant information from the long context, then reasoning over the retrieved information to produce answers. The model is explicitly trained to align with both retrieval and reasoning objectives, leading to improvements over baselines.

**Strengths:**

1. The research shows that breaking down long-context reasoning into explicit retrieval and reasoning steps can enhance LLM performance on long document understanding tasks and also provide more transparent reasoning processes. These combined strengths position the paper with higher value.

2. The model can help with hallucination mitigation, where the system dramatically reduces the tendency to generate fictional content by explicitly aligning the LLM with retrieval objectives.

3. The paper is easy to follow and is well-written.

**Weaknesses:**

1. I believe an important baseline is missing: How is your retrieve-then-reason framework compared to retrieval-augmented generation for LLM baseline methods including DPR (Dense Passage Retrieval), FLARE (Active Retrieval Augmented Generation), RETA-LLM (RETA-LLM: A Retrieval-Augmented Large Language Model Toolkit). Throughout these methods, retrieval can even be performed with a smaller, faster model on a larger context like a database and are applicable to the long-context understanding scenario.

2. The contribution is limited. As the author indicated in sec 4.2, the retrieve-then-reason framework is very similar to "generate quotes and citations first" framework. Therefore, the innovation of this work seems just to fine-tune the models based on these data.

3. There is lack of a clear explanation for some unexpected parts of the experimental results. For example, the model's performance on squad for some lengths (8k, 16k) in Table 1 is significantly lower than shorter and longer lengths, and I have not seen a clear interpretation for this unusual behavior.

**Questions:**

1. As mentioned in the first weakness, can you evaluate both efficiency and effectiveness comparison between these methods, and show how in-context retrieve-then-reason can bring extra benefits compared to RAG?

2. The term "Retrieval" in this paper is different from previous work like DPR, where the models do not retrieve in-context but retrieve on an external codebase. Considering this may cause misunderstanding for unfamiliar readers, I suggest considering changing names like "in-context retrieval"

---

> ### Author Response · Authors · 2024-11-22
> **Response to Reviewer 4gL5**
>
> ### Q1: I believe an important baseline is missing: How is your retrieve-then-reason framework compared to retrieval-augmented generation for LLM baseline methods including DPR (Dense Passage Retrieval), FLARE (Active Retrieval Augmented Generation), RETA-LLM (RETA-LLM: A Retrieval-Augmented Large Language Model Toolkit). Throughout these methods, retrieval can even be performed with a smaller, faster model on a larger context like a database and are applicable to the long-context understanding scenario. (Also see the first comment in Questions)
>
> Following your helpful suggestion, we add the baseline of RAG method for a more comprehensive comparison. We will also cite and discuss the related works you mentioned in our paper.
>
> **RAG Setup**    We employ the DPR model to encode the question and documents.  We use the `facebook-dpr-question_encoder-multiset-base` and `facebook-dpr-ctx_encoder-multiset-base`  encoders provided by `sentence-transformers` to encode the passage and question separately. For each question, we build a retrieval datastore independently based on the in-context documents of it. We retrieve the top five documents according to the dot-product score. Afterwards, we assemble the retrieved documents and the original question using DA template.
>
> **Results**  As shown below, the overall performance of the RAG approach surpasses that of the Direct-Answering (DA) method, although the DA approach outperforms RAG in short-context scenarios. The RAG approach demonstrates extremely stable performance across context lengths ranging from 4K to 128K. However, when using the LLM itself for in-context retrieval, i.e., the Retrieve-then-Reason (RR) method, significant performance gains are observed compared to the RAG approach. This indicates that the LLM has a superior understanding of the relationship between context information and the user’s question.
>
> | Method | 4K | 8K | 16K | 32K | 64K | 128K | Overall |
> | --- | --- | --- | --- | --- | --- | --- | --- |
> | DA | 51.6 | 51.2 | 47.4 | 47.4 | 43.6 | 34.29 | 45.9 |
> | RR | 58.8 | 55.6 | 54.8 | 54.6 | 51.4 | 44.8 | 53.3 |
> | QF | 53.4 | 54.4 | 49.4 | 48.6 | 44.8 | 32.8 | 46.0 |
> | S2A | 43.0 | 44.6 | 42.8 | 40.8 | 34.0 | 30.5 | 39.2 |
> | RAG | 47.4 | 48.4 | 49.2 | 46.8 | 45.6 | 46.6 | 47.3 |
> | ALR$^2$ | 77.2 | 76.8 | 77.2 | 76.6 | 77.5 | 75.2 | 76.7 |
>
> (Experiments Hotpot QA Dataset)
>
> ### Q2: The contribution is limited. As the author indicated in sec 4.2, the retrieve-then-reason framework is very similar to "generate quotes and citations first" framework. Therefore, the innovation of this work seems just to fine-tune the models based on these data.
>
> We agree that the technical part of this work is simple, but the research problem explored in this paper is important and our proposed approach is very effective.
>
> To summarize, we aim to address several critical questions in this work:
>
> 1. **What are the common problems with current long-context LLMs?**  In this preliminary study, we reveal that the long-context capabilities of many commercial LLMs are surprisingly fragile. Specifically, increasing task complexity by requiring additional reasoning significantly degrades the performance of even strong LLMs such as GPT-4, Claude-3, and Command-R. We believe this observation is crucial for the research community.
> 2. **How can the observed problem be solved?**  We address the issue by decomposing the original problem into retrieval and reasoning sub-problems, employing the RAG formulation, i.e., enabling the LLM itself to perform both retrieval and reasoning.
> 3. **What is the impact of this approach on the community?**  Our simple training paradigm significantly stabilizes long-context reasoning performance. Furthermore, we highlight practical challenges, such as severe hallucination, when using general-purpose LLMs for retrieval tasks.
>
> ### Q3: There is lack of a clear explanation for some unexpected parts of the experimental results. For example, the model's performance on squad for some lengths (8k, 16k) in Table 1 is significantly lower than shorter and longer lengths, and I have not seen a clear interpretation for this unusual behavior.
>
> Thanks for the good catch. We hypothesize that this is because the model was trained on a mixed dataset consisting of HotpotQA, SQuAD, and four tasks from NIAH, where SQuAD constitutes only a small portion of the training dataset.
>
> ### Q4: The term "Retrieval" in this paper is different from previous work like DPR, where the models do not retrieve in-context but retrieve on an external codebase. Considering this may cause misunderstanding for unfamiliar readers, I suggest considering changing names like "in-context retrieval”
>
> Thanks for this good suggestion! This is one thing that our authors are also considering. We will clarify the related parts clearer in the revised paper following your suggestion.

---

> > ### Author Response · Authors · 2024-11-27
> > **Additional results regarding your Weakness 1**
> >
> > Since other reviewers are also interested in the performance of RAG approach. In addition to the RAG method using DPR as the retriever, we also explored stronger retriever powered by LLM, i.e., the RankZephyr method [1]. This method ranks the provided passages, and thus we can induce the retrieval results from the rank list.
> >
> > **Setup**: We use the same prompt in RankZephyr for rerank the documents:
> >
> > ```python
> > RERANK_PROMPT = """You are RankLLM, an intelligent assistant that can rank passages based on their relevancy to the query.
> >
> > I will provide you with {num} passages, each indicated by a numerical identifier []. Rank the passages based on their relevance to
> > the search query: {query}
> >
> > {passages}
> >
> > Search Query: {query}
> >
> > Rank the {num} passages above based on their relevance to the search query. All the passages should be included and listed
> > using identifiers, in descending order of relevance. The output format should be [] > [], e.g., [4] > [2]. Only respond with the ranking results, do not say any word or explain."""
> > ```
> >
> > Regarding the representation of each document, we also convert our format `Document N: … {passage} …` to `[N]: … {passage} …`, where N is the id of the passage. We use the top-5 documents introduced from the rank list as the retrieval results and integrate them into the QA prompt.
> >
> > Results:
> >
> > |  | 4K | 8K | 16K | 32K | 64K |
> > | --- | --- | --- | --- | --- | --- |
> > | DA | 51.6 | 51.2 | 47.4 | 47.4 | 43.6 |
> > | RR | 58.8 | 55.6 | 54.8 | 54.6 | 51.4 |
> > | RR /wo LC | 55.6 | 53.0 | 56.0 | 51.2 | 49.8 |
> > | DPR-RAG | 47.4 | 48.4 | 49.2 | 46.8 | 45.6 |
> > | RankZephyr-RAG | 51.9 | 51.2 | 50.5 | 46.8 | 43.4 |
> >
> > We can find that RankZephyr-RAG achieves better performance than DPR-RAG approach. However, both the RR and RR /wo LC methods achieve better performance than RankZephyr-RAG, indicating that the generative retrieval achieves more stable performance in the long-context scenario.
> >
> > We hope the additional results could help you reevaluate our work. We really appreciate your effort on helping us improve this work. If you have additional questions, please feel free to let us know.
> >
> > [1]: RankZephyr: Effective and Robust Zero-Shot Listwise Reranking is a Breeze!

---

### Meta-Review · Area_Chair_jGzi · 2024-12-21

**Metareview:**

This submission introduces ALR2 designed to enhance the long-context reasoning ability of LLMs. While LLMs have larger context windows, the authors argue that their reasoning capability over long contexts often degrades due to difficulty in identifying and reasoning over relevant information. The idea of ALR2 is to address this by designing a retrieve-then-reason framework. First, it retrieves relevant information; Second, it starts to "reason" with related background information. The proposed method is shown to improve retrieval accuracy and reduce hallucination.

The reviewers identified the strengths of this work as:
- this paper is easy to follow and clearly presented (Reviewers 4gL5, Q28w, WCoe).
- the retrieve-then-reason framework can enhances LLM performance on long-context understanding tasks, helping reduce hallucinations (Reviewers 4gL5, Q28w).
- the proposed r-and-r method shows improvements over some baseline models, especially on long-context QA tasks (Reviewers 4gL5, WCoe).

The points that this work could improve upon include the following:
- Several reviewers in the original reviews and discussions mentioned that the experimental design could be improved, particularly on lacks comparisons to important baselines, e.g., DPR, RETA-LLM etc.,  and lacks certain important datasets, and 2) it does not provide fair comparisons due to differences in answer templates and context usage (Reviewers 4gL5, Q28w).
- the experiments lack details, e.g., hyperparameters, training data, ablation studies (Reviewers Q28w, WCoe).
- The idea of using retrieval and reasoning has been presented before in similar methods, and the authors did not offer evidence / arguments to highlight its unique contributions (Reviewers WCoe, Q28w).

During the rebuttal, the authors successfully convinced two reviewers to update their ratings. Relatively speaking, the engagement between reviewers and the authors are enthusiastic, the reviewers were also active during discussion.
- Reviewer WCoe updated the score from 5 to 6. The reviewer also acknowledged that many of the questions were answered by the authors' rebuttal. However, the reviewer remained concerned about its empirical setup and the lack of importance baselines (as mentioned above).
- Reviewer Q28w raised the rating from 3 to 5 as some issues were addressed by the authors during the rebuttal. This reviewer had extensive back-and-forth discussions with the authors regarding the datasets and evaluation metrics used. Ultimately, the reviewer decided  not to increase the score further.
- Reviewer 4gL5 (an initial & final rating of 6) did not respond to the authors' rebuttal.


Overall, the final ratings for this work are 6, 6 (up from 5), 5 (up from 3). The reviewers' main concerns on technical novelty, empirical demonstration (missing baselines and important datasets / metrics) remained unaddressed after the rebuttal. From the AC's standpoint, the idea of retrieval-and-reasoning presented in this work does not outweigh these issues. Based on these factors, this work in its current form is not recommended for acceptance, since it would significantly benefit from another round of revision.

**Additional Comments On Reviewer Discussion:**

During the rebuttal, the authors successfully convinced two reviewers to update their ratings. Relatively speaking, the engagement between reviewers and the authors are enthusiastic, the reviewers were also active during discussion.
- Reviewer WCoe updated the score from 5 to 6. The reviewer also acknowledged that many of the questions were answered by the authors' rebuttal. However, the reviewer remained concerned about its empirical setup and the lack of importance baselines (as mentioned above).
- Reviewer Q28w raised the rating from 3 to 5 as some issues were addressed by the authors during the rebuttal. This reviewer had extensive back-and-forth discussions with the authors regarding the datasets and evaluation metrics used. Ultimately, the reviewer decided  not to increase the score further.
- Reviewer 4gL5 (an initial & final rating of 6) did not respond to the authors' rebuttal.

---

### Decision · Program_Chairs · 2025-01-22

Reject